# The Low Survivability of Transplanted Gonadal Grafts: The Impact of Cryopreservation and Transplantation Conditions on Mitochondrial Function

**DOI:** 10.3390/biology13070542

**Published:** 2024-07-18

**Authors:** Inês Moniz, Maria Soares, Ana Paula Sousa, João Ramalho-Santos, Ana Branco

**Affiliations:** 1Doctoral Programme in Experimental Biology and Biomedicine (PDBEB), Institute for Interdisciplinary Research, University of Coimbra, 3030-789 Coimbra, Portugal; inescmoniz@hotmail.com (I.M.);; 2CNC—Centre for Neuroscience and Cell Biology, CIBB—Centre for Innovative Biomedicine and Biotechnology, University of Coimbra, Azinhaga de Santa Comba, Polo 3, 3000-548 Coimbra, Portugal; 3Reproductive Medicine Unit, Unidade Local de Saúde de Coimbra, Praceta Prof. Mota Pinto, 3000-075 Coimbra, Portugal; 4Eugin Coimbra, Rua Filipe Hodart, 3000-185 Coimbra, Portugal; 5Department of Live Sciences, University of Coimbra, Calçada Martim de Freitas, 3000-456 Coimbra, Portugal

**Keywords:** oncofertility, cryopreservation, transplantation, mitochondrial dysfunction, oxidative stress, fertility preservation

## Abstract

**Simple Summary:**

Gonadal tissue transplantation as a fertility preservation technique is highly conditioned by our current knowledge of functionality following transplantation, tissue cryodamage, and ischemia–reperfusion injury. This paper presents an updated review of the literature on mitochondrial dysfunction and oxidative stress in the context of gonadal tissue cryopreservation and transplantation.

**Abstract:**

Advances in tissue preservation techniques have allowed reproductive medicine and assisted reproductive technologies (ARTs) to flourish in recent years. Because radio- and chemotherapy procedures are often gonadotoxic, irreversible damage can preclude future gamete production and endocrine support. Accordingly, in recent years, the freezing and storage of gonadal tissue fragments prior to the first oncological treatment appointment and autologous transplantation post-recovery have been considered improved solutions for fertility recovery in cancer survivors. Nevertheless, the cryopreservation and transplantation of thawed tissues is still very limited, and positive outcomes are relatively low. This review aims to discuss the limitations of oncofertility protocols with a focus on the impacts of mitochondrial dysfunction, oxidative stress, and the loss of antioxidant defense in graft integrity.

## 1. Fertility Preservation in Oncological Patients

Fertility preservation has evolved considerably in the past two decades. Young patients with cancer undergoing radiotherapy, systemic chemotherapy, and/or oncological gonadectomy can have their fertility severely compromised, as most of these therapeutic approaches target not only malignant cells, but also a wide variety of healthy cells. With chemotherapy, the loss of reproductive function can occur in a cell-cycle-dependent manner—with the usage of antimetabolites (e.g., Cytarabine)—or in a cell-cycle-independent manner—with anthracyclins, alkylating, and/or platinum agents [1,2,3]. These highly aggressive drugs can have short- and long-term effects on the ovary, starting with primordial follicle atresia and apoptosis and culminating in inflammation and vascular and stromal injury, permanently disrupting estradiol production and altering the follicular reserve [1,3]. In the testis, these compounds can damage the germinal epithelium and target Leydig and proliferating spermatogonial stem cells (SSCs), disrupting sperm and hormone production and ultimately contributing to infertility [4,5].

Radiotherapy to the head and neck is potentially damaging to the hypothalamus–pituitary axis and to the central nervous system (hindering gonadotropin production), whereas exposure to the pelvic and abdominal regions can directly damage the highly sensitive testes and ovaries [6,7].

In a 2018 survey, more than 60% of all cancer patients monitored reported a strong desire to have children [8], while a 2023 survey involving pediatric patients found that more than 50% of all questionees wished to have children in the future [9]. Therefore, all patients should be counseled about the possible risks of infertility before beginning oncological treatment to guarantee full awareness of all available options.

For post-pubertal men, sperm cryopreservation and banking have been viable options for several decades. Nevertheless, these approaches are unviable for non-sperm-producing pre-pubertal patients with cancer [10]. In such cases, there have been ongoing clinical trials focused on the possibility of cryopreserving testicular tissue (TT) containing SSCs. This experimental protocol encompasses a unilateral orchiectomy or an open testicular biopsy to collect the testicular tissue and carry out its preparation and fragmentation, the embedding in cryoprotectant solution, and its ensuing cryopreservation. After completing all oncological treatments, the preserved tissue is thawed and grafted to homotopic or ectopic sites (e.g., peritoneal space) [5,11,12]. This provides cancer survivors with a tool to recover hormone production and restore spermatogenesis. Fayomi and colleagues recently proved that the autologous transplantation of cryopreserved testicular tissue into a priorly damaged (through chemotherapy) testis could recover the endocrine function and fertility of non-human primates [13]. As of 2023, the cryopreservation of testicular tissue from pre- and post-pubertal patients has been reported in more than 700 patients worldwide [10,11,12,13,14]. Thus, although still mostly experimental, testicular tissue cryopreservation might be a possible method of preserving germ cell lines in pre-pubertal patients [11,12,15].

Contrastingly, the cryopreservation of unfertilized mature oocytes is the most established fertility preservation option for adult women [2,10]. This method allows patients to preserve oocytes in the absence of a reproductive partner, proving an advantage over the cryopreservation of fertilized eggs [2]. However, to complete fertilization, this option requires intracytoplasmic sperm injection (ICSI) and follicular phase controlled ovarian stimulation, which is inconceivable in young patients and many times impractical in adults due to the urgency to begin oncological treatments. Most importantly, oocyte cryopreservation fails to legitimately restore reproductive independence and ovarian and endocrine function to cancer survivors [2]. Instead, patients can opt to cryopreserve ovarian tissue grafts for future autologous transplantation. This protocol encompasses a surgical laparoscopy or laparotomy, followed by the segmentation of the collected tissue into small transplantable fragments, the equilibration of the tissue in a cryoprotectant solution, and, finally, the cryopreservation of the prepared grafts by slow-freezing or vitrification [16,17,18]. Contrary to testicular tissue cryopreservation, ovarian tissue banking is no longer experimental, and since the first human birth registered in 2004, it has allowed patients—even those unqualified for ovarian stimulation and IVF—to successfully preserve thousands of immature follicles [17,18,19]. After completing all oncological treatments, the tissues are thawed, cleansed of their cryoprotective media, and transplanted back into the remaining ovary, eventually restoring the patient’s fertility and endocrine function. A 2022 meta-analysis that included 735 women throughout 87 distinct studies estimated that this fertility preservation technique summed a pooled success rate of 37% for pregnancies and 28% for life births [13,20]. Endocrine function—indicated by the production of estrogen, follicle-stimulating hormone (FSH), luteinizing hormone (LH), and anti-Müllerian hormone (AMH)—was recovered in most of the cases, validating this protocol as a reasonably reliable tool for cancer survivors [20,21].

Regardless, pregnancy and live birth rates continue to be subpar in patients who undergo these procedures, so it is important to notice that the cryopreservation of human samples for future clinical application—whether cellular or tissue in nature—is still highly conditioned by our current understanding of the mechanisms behind cryodamage and ischemia–reperfusion (I/R) injury.

## 2. Cryopreservation: Principles of Cryopreservation and Known Limitations

Cryopreservation is the process whereby living cells, tissues, or organs can be stored for an indeterminate amount of time in sub-zero temperatures while maintaining structure, viability, and biological function in a transiently quiescent state [22,23]. Most cryopreservation techniques follow the same core principles, starting with bathing the tissue in a cryoprotectant (CPA) solution, followed by slow-freezing or vitrification, and ending with storage in liquid nitrogen [24]. This first step is essential for the successful banking of biological samples, as it impedes ice crystal formation and physical and osmotic injury that disrupts cellular membranes and intracellular structures [23,25]. Most cryoprotectant solutions can be categorized into permeating CPAs—which include glycerol, dimethyl sulfoxide (DMSO), ethylene glycol (EG), and 1,2-propanediol (PROH)—and non-permeating CPAs—such as sugars (e.g., sucrose and trehalose) and proteins (e.g., human serum albumin (HSA)) [26,27,28,29].

The preservation of complex tissues can be an especially arduous process. Whereas with single-cell-line cryopreservation, the main obstacle is to maintain the cell’s internal architecture and to suspend all biological function, with complex tissues—many times comprising different cell lines and organized structures and vessels—there is a need to uniformly preserve each cell type, their three-dimensional organization, and biochemical activity, while also taking into account that cells from different natures might respond differently to specific CPAs and cryogenic temperatures [23,30,31]. Diffusion through multi-cellular structures can be hindered by differing membrane permeabilities and complexities; ergo, each cryopreservation protocol must be designed and optimized according to the particularities of the target tissue [23,30]. Moreover, composite tissues are also susceptible to damage caused by the formation of ice crystals and osmotic injury.

Cryoprotectants can, for the most part, prevent cryodamage by limiting water transport and by stopping ice crystal formation [26,27,28]. However, CPAs can also contribute to cell toxicity during cryopreservation [28,32,33]. Thus, recent efforts have focused on exploring new techniques that might mitigate the damage caused by the freezing process and by the cytotoxicity inherent to the usage of CPAs [34,35].

### 2.1. Cryodamage: Mitochondrial Dysfunction and Oxidative Stress

Under homeostatic conditions, mitochondria generate adenosine triphosphate (ATP) via oxidative phosphorylation (OXPHOS), with the passage of electrons being carried out through the electron transport chain (ETC); at the final complex, the electrons are delivered from cytochrome c oxidase to molecular oxygen (O_2_), reducing it to water (H_2_O) [36,37,38]. Throughout this process, the leak of electrons from the ETC (0.1–2% of the total electron transport) can, instead, precociously reduce molecular O_2_ to the free radical superoxide (O_2_^•−^) that, in turn, can be converted into hydrogen peroxide (H_2_O_2_) by superoxide dismutases (SODs) [36,37,38]. In normal physiological levels, O_2_^•−^ and H_2_O_2_ can participate in numerous redox pathways to eliminate invading microorganisms [39,40]. Under cryogenic temperatures, however, ROS production can escalate to dangerous levels (Figure 1).

Throughout the freezing protocol, the cell enters a metabolically dormant state, where mitochondrial ATP is not produced. During this time, the mitochondria can undergo a permeability transition, resulting in swelling that, in extreme cases, culminates in the rupture of the mitochondrial membrane [41,42]. The consequent release of proteins from ruptured organelles activates caspases and nucleases to initiate apoptosis. In fact, several reports reveal that cryopreserved samples can suffer mitochondrial dysfunction, concurrent with a great decrease in ATP synthesis and a surge in the generation of mitochondrial ROS (mtROS) [42,43,44,45]. Excess O_2_^•−^ drives the export of cytochrome c from the mitochondria into the cytosol, where it binds to Apaf-1 to form apoptosomes that, in turn, activate caspase-9 to trigger apoptosis [37,45,46]. Likewise, surplus H_2_O_2_ can undergo the Fenton reaction—triggered by Fe(II) salts—to generate the highly reactive hydroxyl (•OH); this free radical can cleave the covalent bounds in proteins and carbohydrates to induce DNA damage and disrupt chromosomal alignment and microtubule formation in metaphase II mouse oocytes [40,45,46].

It is important to note that the freezing–thawing cycle also strongly reduces SOD activity and the expression of intracellular glutathione (GSH) [47,48]. While SOD catalyzes the removal of O_2_^•−^, GSH is responsible for degrading H_2_O_2_ [40]. Thus, by lowering SOD and GSH activity, cryopreservation compromises the cells’ antioxidant defenses and leaves them vulnerable to further oxidative damage.

CPAs fail to protect the cell against oxidative stress and, alone, can be a source of ROS. For instance, DMSO is known to illicit endoplasmic reticulum (ER) stress that results in the exacerbated release of Ca^2+^ from the ER. The excess ions are then incorporated by the mitochondria, where they prompt mitochondrial Ca^2+^ overload, swelling, and loss of mitochondrial membrane potential (MMP). In concert, these alterations magnify the generation of ROS in a redundant and vicious cycle [49,50]. Ultimately, there is still a need to develop new strategies that mitigate mitochondrial dysfunction and oxidative damage caused by the currently established cryopreservation techniques.

### 2.2. Freezing Conditions and the Mitochondrial Health of Reproductive Tissues

With ovarian tissue cryopreservation (OTC), follicle loss begins with the mechanical damage intrinsic to the initial preparation of the tissue. After the ovary is collected, the ovarian medulla is removed so that only a thin section of the cortex remains. This fragmentation facilitates the rapid penetration of the CPAs into the tissue, reducing the formation of ice crystals and effectively minimizing the harm caused by low temperatures. Naturally, during this thinning process, the largest developing follicles, including most of the primary, secondary, and antral follicles, are lost [49]. Following preparation, the tissue is exposed to second and third aggressors, the cytotoxic CPAs, and sub-zero temperatures. Historically, the first report of a human life birth after OTC used DMSO (1.5 mmol/L) and followed a slow-freezing protocol [19]. Most techniques now employ a combination of permeating and non-permeating CPAs. Additionally, given the significant stromal cell damage and follicle loss observed during most slow-freezing protocols, vitrification has emerged as a promising alternative to store OT. This approach involves ultra-swift cooling in the presence of concentrated CPAs, inducing an ice-free, glassy, amorphous state that effectively preserves stromal integrity and safeguards the sample against cellular damage. Vitrification assures a more efficient alternative to slow-freezing, particularly in regard to the preservation of secondary follicles and stromal cells, as it minimizes the damage caused to granulosa cells in secondary follicles (Table 1) [50,51].

It has been hypothesized that primordial follicles are the most resistant to cryogenic conditions since they have lowered metabolic activity. Studies carried out on different animal models confirmed that the mitochondria of oocytes contained within cryopreserved OT exhibited ultrastructural abnormalities, including the loss of cristae, granulated matrixes, swelling, and degeneration [61,62,63]. Simultaneously, the neighboring stromal cells presented loss of intracellular organelles, vacuolization, and similar mitochondrial abnormalities [62,63]. Other studies identified a significant loss of MMP and a decrease in mitochondrial OXPHOS in oocytes and stromal cells from cryopreserved-thawed OT, respectively [62,64]. In OT grafts, mitochondrial dysfunction is progressively more evident in its inner sections, closer to where the medulla was removed, rather than the outer sections [21]. Coincidently, follicle loss during OTC protocols is more marked in the medullar area [16,21]. In a recent study, Wu and colleagues reported that transplanted cryopreserved OT suffered a higher decline in follicle reserve when compared to transplanted fresh OT [64]. These results are consistent with those obtained by Rodrigues et al., who found that cryopreservation markedly decreased the tissues’ global oxygen consumption rates and the number of morphologically intact follicles—especially in tissues undergoing transplantation after being previously cryopreserved (transplanted-cryopreserved) [65]. Conjointly, these studies confirm that OTC directly impacts the graft’s ability to survive transplantation with its initial follicle pool intact, probably due to a decline in mitochondrial function and a reduced ability to meet the required energetic demands to survive transplantation. As described in the previous section, in frozen samples, mitochondrial dysfunction is accompanied by a corresponding increase in the generation of ROS. Studies show that slow frozen OT displays abnormally high levels of ROS, known to interact and damage lipids, proteins, and nucleic acids, prompting phospholipidic membrane disruption and DNA double-strand breakage [58,64,65]. Such alterations potentiate telomere shortening and the occurrence of spindle malformations and chromosomal abnormalities that not only contribute to tissue injury and stromal death, but also to a decline in oocyte quality [58,64,65] (Figure 1).

As a mostly experimental procedure, there is still no consensus regarding the freezing conditions, the type and concentration of CPAs used, or the thawing/warming protocols followed when cryopreserving testicular tissue. The few trials conducted with humans used a combination of permeating and non-permeating CPAs at varying concentrations to minimize individual cytotoxicity (Table 2).

Nevertheless, these few studies demonstrated that these storing conditions contribute to structural damage to the seminiferous tubes and ultrastructural changes to spermatogonia and SCs, as evidenced by increased vacuolization and dilated mitochondrial cristae [5,66]. In animal models, different reports also attest to mitochondrial damage connected to a wide variety of testicular tissue cryopreservation (TTC) conditions. In fact, studies demonstrated that most CPAs promote the formation of free radicals and lipid peroxides [71,72]. As with OTC, TT slow-freezing protocols lead to intracellular ice crystal formation and high levels of ROS that damage sperm cell integrity, membranes, DNA structure, and mitochondrial function [73]. After thawing, TT samples can also exhibit germ cell DNA fragmentation, mitochondrial dysfunction, a lowered MMP, and increased ROS levels concurrent to a decrease in cell viability, cytolysis, and tissue disintegration [71,74,75,76]. Moreover, by severely decreasing the GSH content and SOD and CAT activity, TTC also contributes to the loss of antioxidant capacity, increasing the potential damage to these structures [75,77].

## 3. The Transplantation of Cryopreserved Gonadal Tissues: The Effects of Ischemia/Reperfusion (I/R) on Reproductive Cells, Mitochondrial Function, and Tissue Integrity

Although cryopreservation damages gonadal tissues, it is important to stress that mass gamete loss occurs primarily after the transplantation of the preserved graft. Indeed, several studies report that follicle loss in ovarian grafts occurs essentially after autotransplantation [49,78]. Some estimate that transplantation is responsible for the loss of 26% of the primordial follicle reserve and up to 75% of all follicle loss during these fertility preservation protocols; this is mostly attributed to I/R injury (IRI) [49,52].

The ovarian graft is first exposed to ischemia when it is removed from the ovary and separated from its original nutrient and oxygen supply. Without vascular anastomosis, the tissue remains ischemic since angiogenesis and revascularization are only complete after an average of 10 days [78]. During this ischemic period, the OT is susceptible to hypoxic damage. Deprived of O_2_, ATP production through OXPHOs is interrupted, and a more glycolytic anaerobic metabolism is induced. This causes the ATP levels to plummet and results in the reduction in redox-active enzymes and ETC carriers (e.g., NADH and coenzyme Q) [79,80]. Hypoxic cells are unable to meet the energetic demands required to keep the tissue healthy, although primordial follicles are marginally more resistant to hypoxic injury and can remain intact even when surrounded by damaged cortical stromal cells [81]. Nevertheless, hypoxic injury triggers the production of ROS and the release of cytokines and growth factors (e.g., TGF-β and TNF-α) that promote apoptosis and the recruitment of immune cells to initiate an inflammatory response [82]. Meanwhile, there is a build-up of metabolic products (such as lactate and succinate)—which cause metabolic acidosis—and a rise in the production of hypoxanthine and lipid peroxides [83,84,85,86]. With revascularization, the reestablishment of blood vessels sends a rush of said products to the tissue; succinate is quickly oxidized, causing the mass production of O_2_^•−^ via reverse electron transport, whereas xanthine oxidase catalyzes the oxidation of the produced hypoxanthine to xanthine in a reaction that reduces O_2_ to form O_2_^•−^ and H_2_O_2_ [79,84,85,87]. This burst of ROS causes damage to proteins and nucleic acids and leads to the peroxidation of lipids that disrupts cell membrane integrity; these ROS are also responsible for the activation of the mitochondrial permeability transition pore (MPTP), which allows for the release of mitochondrial apoptotic and inflammatory cytokines—such as cytochrome c and TNF-α, respectively—which trigger apoptotic and inflammatory pathways [80,83,85,88,89].

Regardless, primordial follicle loss post-transplantation is not exclusively caused by cell death and tissue injury. Numerous reports confirm that after surgery, a great number of inactive follicles undergo premature activation as early as 3 days post-transplantation. With ischemia and oxidative stress, there is a corresponding release of insulin-like growth factor-1 (IGF-1) and hypoxia-inducible factor-1 (HIF-1) that stimulate the rise in granulosa cell proliferation and follicle growth, concurrent to early primordial follicle activation [50,89,90]. This “follicle burnout” depletes the pool of dormant follicles and compromises the long-term reproductive function of the patient [91]. In concert, ischemia and reperfusion provoke mitochondrial alterations that can be directly connected to OT inflammation, degeneration, edema, vascular congestion, and early follicle activation, and can thus be considered greatly responsible for the decline in ovarian reserve observed in transplanted ovarian grafts (Figure 2) [80].

Similar results have been reported in the limited studies focused on TT transplantation. Accordingly, transplantation, rather than cryopreservation, is also to blame for the significant loss of spermatogonia observed in grafted TT. After the removal of the testicular strips from the testis, and during the ischemic period before revascularization, ATP production through the ETC declines, and the levels of hypoxanthine rise [92]. This is consistent with a recent report that attested to a significant loss of SSCs during the ischemic period, concurrent with tubule degeneration at the center of the graft [12]. The angiogenesis of testicular grafts can be completed within 7 days after transplantation; the reestablishment of the blood supply to the transplant causes a burst in ROS production that increases oxidative stress. This, in turn, activates mitogen-activated protein kinases and furthers lipid peroxidation, protein and DNA damage, and the initiation of apoptotic pathways [92,93].

## 4. Prevention against Cryodamage

Multiple complementary approaches involve enhancing cryoprotectant solutions with supplements such as antioxidants, hormones, and anti-apoptotic agents. For instance, the combination of N-acetylcysteine (NAC) and pulsed electromagnetic fields (PEMFs) in vitrified mouse ovarian tissue demonstrated a synergistic effect in promoting angiogenesis and protecting the OT against oxidative stress and inflammation [94]. On the other hand, antioxidants such as vitamin E or selenium have been shown to mitigate oxidative stress during the freezing and thawing processes. In ducks, the administration of the antioxidant resveratrol, before cryopreservation, significantly upregulated the expressions of the VEGF, HIF-1α, Nrf2, CAT, and Bcl-2 mRNA genes, which are associated with increased antioxidant capacity (via SOD, CAT, and GSH peroxidase upregulation), decreased apoptosis, and enhanced angiogenesis post-transplantation [95].

Alternatively, pre-conditioning techniques such as ischemic preconditioning or hormonal priming can be applied to augment graft survival and function in the subsequent post-transplantation phase. The combined use of the hormone FSH and the anti-apoptotic agent S1P during vitrification has been shown to preserve the primordial follicle pool and suppress apoptosis [96]. The use of the anti-freezing protein during the warming procedure can also prevent OT damage and improve ovarian follicle morphology and apoptosis [97] (Table 3).

With testicular tissue, most studies have addressed the need for optimizing CPA solutions and concentrations. DMSO is considered more effective for immature TT—which is more susceptible to CPA toxicity—whereas EG has been indicated for adult TT more [30]. Supplementation with trehalose has also been consistently reported to harbor positive results when it comes to reducing the number of apoptotic cells—via BAX downregulation and BCL-2 upregulation—and preserving testosterone production by cryopreserved Leydig cells [75,102,103,104]. This compound is known to promote the activity of SOD and CAT, antioxidant enzymes that protect cells against cryogenic-associated oxidative damage [75]. By contrast, vitamin A was reported to preserve SSC differentiation capacity and global cell division in TT while preventing spermatid DNA damage and nuclear alterations, ultimately promoting a higher spermatic yield [105] (Table 4; Figure 3).

## 5. Prevention against Ischemic Damage in Transplanted Gonadal Grafts

As previously discussed, the primary challenge encountered during gonadal tissue transplantation is ischemia and ischemia–reperfusion injury. Therefore, efforts have been primarily focused on reducing the damage caused by inadequate blood supply rather than on solely preventing cryoinjury.

In mice, the administration of hormones such as erythropoietin increased angiogenesis, reduced ischemic damage, decreased fibrosis, and maintained ovarian follicle proliferation after transplantation [106]. Taurine, on the other hand, was found to decrease oxidative stress and apoptosis and expedite angiogenesis by augmenting CD31 expression (Figure 4) [107]. In this context, the pro-angiogenic growth factor VEGF also has a prominent role in follicle growth as it promotes endothelial proliferation and triggers signaling pathways crucial for early follicle activation. These pathways include PI3K/Akt and the mammalian target of rapamycin (mTOR) [108]. Hence, as master regulators of primordial follicle recruitment, the pharmacological inhibition of PI3K—using LY294002—or the suppression of mTOR—using rapamycin or the NGF inhibitor K252a—was revealed to safeguard the tissue against follicle pool depletion and overexhaustion by effectively preventing the premature activation of primordial follicles from their dormant state [56,101,109,110,111,112]. This renders the tissue more suitable for clinical application. Moreover, treatments based on the recombinant anti-mullerian hormone (which acts as an inhibitor of primordial follicle activation) have also demonstrated efficacy in preserving the primordial follicle pool by suppressing their transition to primary follicles [56].

As reoxygenation kinetics exacerbate oxidative damage, new strategies are also being developed with the goal of enhancing revascularization and limiting the prolonged exposure to ischemia. The daily administration of NAC (up to 7–12 days after transplantation) produced promising results by reducing IRI and promoting follicle survival in immunodeficient mice; these are effects that were attributed to the upregulation of the antioxidant defense system (increased SOD1, HMOX1, and CAT) and to a boost in anti-inflammatory and antiapoptotic mechanisms (via BCL2) [113]. The combination of NAC and estradiol (E2) has produced similar outcomes [114]. The administration of glutathione (GSH) and ulinastatin (UTI), on the other hand, has helped block macrophage accumulation and increase the levels of VEGF, CD31, and SOD 1 and 2 in mice, who also presented lowered levels of IL6, TNF-α, and MDA [115]. Thus, various strategies aimed at mitigating inflammation and oxidative harm have been explored; these also include the use of L-carnitine and Etanercept, the latter of which is known to neutralize the activity of TNF-α and inhibit the release of cytokines, chemokines, and stress hormones [116,117].

Finally, adult stem cells exhibit suitable attributes when it comes to maintaining the follicle pool. They are considered promising tools for tissue regeneration and possess the ability to release paracrine factors that, in turn, modulate a wide variety of cellular mechanisms from apoptosis to inflammation [118,119]. Ergo, some authors have presented the usage of mesenchymal stem cells (MSCs) as a tool to protect against ischemic exposure [120,121,122]. Adipose stem cells (ASCs) were observed to encompass and penetrate OT grafts and increase vessel density and angiogenesis through the secretion of VEGF [123]. The resulting increase in angiogenesis is pivotal in elevating follicle survival rates and reducing apoptosis and follicular activation. One study demonstrates that oogonial stem cells (OSCs) can also be incorporated by the OT and partially restore normal ovarian function after chemotherapy; by maintaining the ability to differentiate into oocytes, these cells were successfully matured to produce a viable offspring through in vitro fertilization [124] (Table 5).

Notably, MSCs also demonstrate a promising ability to produce large quantities of extracellular vesicles (EVs)—lipidic bilayer complexes capable of transporting cargo such as microRNAs (miRNA), small interfering RNA (siRNA), messenger RNA (mRNA), proteins, and lipids [126]. Due to their low immunogenicity, membrane integrity, and stability, EVs have been considered important cell-free therapeutic tools for translational applications [127]. Studies carried out in the context of organ and tissue transplantation have revealed that these EVs can be endocytosed and significantly reduce and attenuate inflammation and I/R damage [128,129,130]. Interestingly, recent reports demonstrate that these cells can also transfer viable mitochondria to damaged neighboring cells, thereby contributing to tissue regeneration [131,132]. Altogether, these reports provide the theoretical basis to support the idea that adult stem cells and EVs might be useful solutions to minimize IRI caused to gonadal grafted tissues, and that their usage and further research on them might be of relevant scientific and clinical interest.

While numerous other studies have demonstrated the efficacy of various compounds in enhancing ovarian cortex transplantation programs, there is still comparatively little research on testicular tissue transplantation. Nevertheless, some authors have ventured into similar strategies adapted to TTC. These include supplementation with diverse compounds, namely with gonadotropin (FSH/LH) and Knockout Serum Replacement (KSR) [133]. Concomitantly, Del Vento and colleagues were able to improve TT revascularization by encapsulating the grafts in calcium alginate hydrogels and delivering VEGF and platelet-derived growth factor (PDGF)-loaded polymeric nanoparticles [134].

In summary, strategies aimed at promoting early blood supply to the graft, mitigating germ cell burnout post-transplantation, and reinforcing the tissue’s antioxidant, anti-apoptotic, and anti-inflammatory capacities have consistently been the primary focus of most studies. Although mitochondrial dysfunction can be seen at the basis of all these comorbidities, there is very little research targeting and prioritizing mitochondrial dysfunction and metabolic alterations in cryopreserved transplanted gonadal grafts.

## 6. Conclusions and Future Perspectives

With the rise in cancer survivors intrinsic to improved survival odds, there has been an accompanying demand for ART, and the cryopreservation of gonadal tissues has been increasingly carried out as a resort. There is still limited research focused on the impact that these protocols have on the energetic and biomolecular functions of reproductive structures and cells; ergo, to understand the shortfalls of fertility preservation techniques, scrutiny should go not only to the systemic, quantitative, and morphological alterations caused by cryostorage and transplantation, but also further into the ultrastructural, bioenergetic, and metabolic abnormalities caused by these techniques.

A lot of recent research has ventured into medium supplementation to mitigate cryodamage and IRI. Apropos of cryostorage, a combination of different permeating and non-permeating agents—namely DMSO and trehalose—stands out as a promising method to safeguard germ cell integrity. MSCs—with their low immunogenicity and ability to release protective EVs, healthy mitochondria, and anti-inflammatory, anti-apoptotic, and antioxidant factors—might also be a future tool for the prevention against mitochondrial dysfunction-associated tissue degeneration.

How these preventive measures safeguard mitochondrial function and oxidative and energetic balance remains to be confirmed and merits further research. We anticipate that by targeting the mitochondria, we might find an important solution for the protection of gonadal tissues against cryodamage and IRI.

## Figures and Tables

**Figure 1 biology-13-00542-f001:**
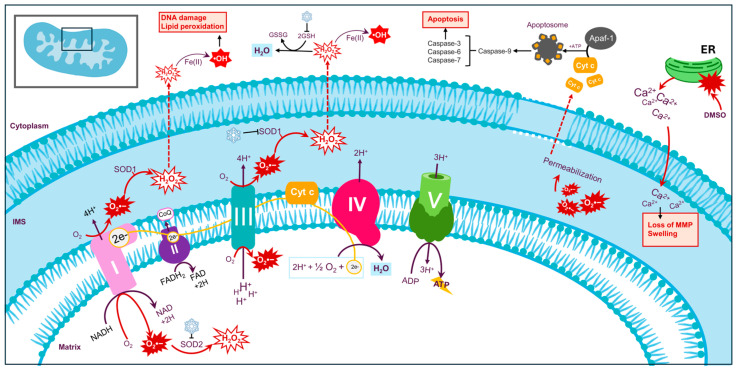
The generation of mitochondrial ROS under cryogenic conditions. The electrons (e-) are transported through the ETC (complex I, complex II, coenzyme Q (CoQ), complex III, cytochrome c (Cyt c), and complex IV) to ultimately produce ATP (complex V). Under cryogenic conditions, the leak of electrons from the ETC reduces O_2_ to O_2_^•−^. Excess O_2_^•−^ alters the mitochondrial membrane’s permeability, resulting in the release of cytochrome c (Cyt c) to the cytosol. Cyt c interacts with Apaf-1 to form the apoptosome that, in turn, activates caspase-9 to trigger the caspase cascade and initiate apoptosis. O_2_^•−^ is converted to H_2_O_2_ by superoxide dismutase 1 (SOD1) on the mitochondrial intermembrane space (IMS) and by superoxide dismutase 2 (SOD2) on the mitochondrial matrix. Under cryogenic conditions, the activity of both SODs is inhibited. H_2_O_2_ can traverse to the outer mitochondrial membrane, where it is converted to the highly damaging •OH. GSH fails to degrade H_2_O_2_ in cryogenic conditions. DMSO elicits ER stress, resulting in the mass release of Ca^+^. The excessive internalization of Ca^+^ by the mitochondria results in mitochondrial swelling and the loss of MMP. Abbreviations: NAD (nicotinamide adenine dinucleotide); FAD (flavin adenine dinucleotide); and GSSG (glutathione disulfide).

**Figure 2 biology-13-00542-f002:**
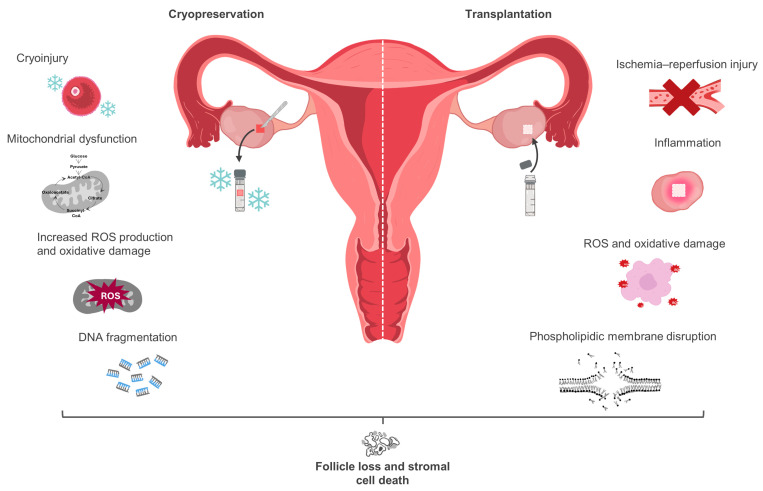
Cryopreservation and transplantation damage to ovarian tissue.

**Figure 3 biology-13-00542-f003:**
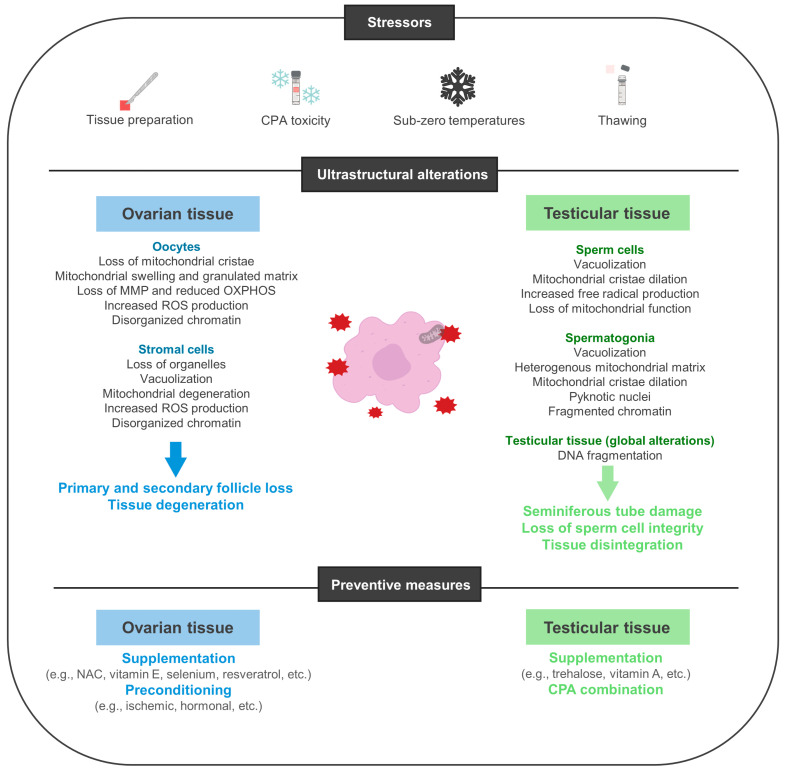
Gonadal tissue cryodamage and preventive measures.

**Figure 4 biology-13-00542-f004:**
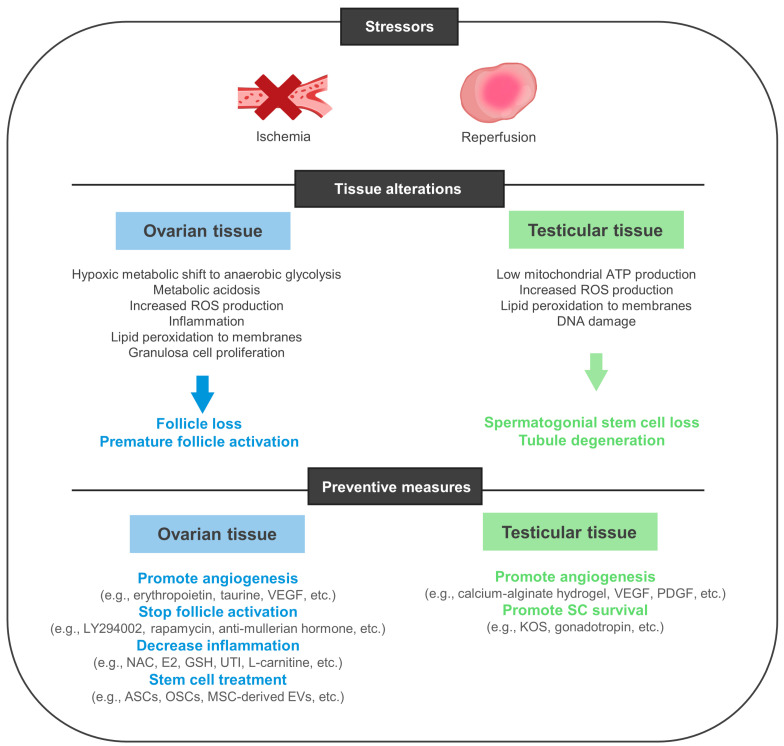
Prevention against ischemic damage in transplanted gonadal grafts.

**Table 1 biology-13-00542-t001:** Most used protocols and CPA concentrations for human OT cryopreservation.

Freezing Technique	CPA	Working Concentration	Reference
Slow-freezing	DMSO	0.0015–1.5 M	[19,50,52,53]
	EG	1.5 M	[32,52]
	PROH	1.26–1.5 M	[21,52,54,55]
	Glycerol	1.5 M	[32,52]
	Sucrose	0.1–0.175 M	[21,54]
Vitrification	DMSO	2 M; 20%	[56,57]
	EG	17–38%	[56,58,59,60]
	Trehalose	0.2–0.5 M	[57,59]
	Sucrose	0.175–1 M	[55,56,58,60]

**Table 2 biology-13-00542-t002:** Most used protocols and CPA concentrations for human TT cryopreservation.

Freezing Technique	CPA	Working Concentration	Reference
Slow-freezing	DMSO	0.7 M; 5%	[5,11,66,67]
	EG	1.5 M	[68,69]
	PROH	1.5 M	[5]
	HSA	5%; 10 mg/mL	[11,66]
	Glycerol	6%	[5]
	Sucrose	0.1 M	[5,67,68,69]
Vitrification	DMSO	2.8 M; 15%	[67,70]
	EG	2.8 M; 15%	[67,70]
	HSA	25 mg/mL	[67,70]
	Sucrose	0.5 M	[67]

**Table 3 biology-13-00542-t003:** Methods proposed to mitigate cryodamage to OT fragments.

Author, Date	Model	Treatment	Main Findings
Bedaiwy et al., 2006 [98]	Human	Slow-freezing; intact ovary with vascular pedicle; DMSO	75% and 78% primordial follicle viability
Westphal et al., 2017 [99]	Bovine, human	Slow-freezing; perfusion and submersion in DMSO	90–100% protection against cryodamage
Lee et al., 2021 [100]	Human	Slow-freezing; Z-VAD-FMK	Improved follicle preservation Improved follicular cell proliferation Prevention against DNA damage
Terren et al., 2021 [101]	Mouse	Slow-freezing; rapamycin and LY294002	Preservation of primordial follicle reserve
Kong et al., 2021 [97]	Bovine	Vitrification; anti-freezing protein (AFP)	↑ OT quality after xenotransplant Prevention against OT damage and apoptosis Improvement in follicle morphology
Rasaeifar et al., 2023 [94]	Mouse	Vitrification; NAC and PEMF	↑ angiogenesis Protection against oxidative stress Protection against inflammation
Wang et al., 2023 [96]	Mouse	Vitrification; FSH and S1P	Preservation of the primordial follicle pool ↓ follicular atresia Suppression of cell apoptosis
Qin et al., 2023 [95]	Duck	Vitrification; resveratrol	↑ VEGF, HIF-1α, Nrf2, CAT, and Bcl-2 mRNA expression ↓ TUNEL-positive cells

↑ Increase; ↓ Decrease.

**Table 4 biology-13-00542-t004:** Methods proposed to mitigate cryodamage to TT fragments.

Author, Date	Model	Treatment	Main Findings
Zhang et al., 2015 [75]	Bovine	Slow-freezing; trehalose	↑ Viability ↑ Antioxidant enzyme activity (SOD and CAT) ↓ Oxidative damage
Dumont et al., 2016 [105]	Mouse	Vitrification; vitamin A	↑ Tissue development Improved differentiation of SSCs ↑ Cell division ↓ DNA damage ↓ Round spermatid nuclear alterations
Xi et al., 2019 [102]	Goat	Slow-freezing; trehalose	↓ Apoptosis Downregulation of BAX Upregulation of BCL-2, CREM, BOULE and HSP70-2 ↑ Testosterone production by Leydig cells
Jung et al., 2020 [103]	Monkey	Slow-freezing; trehalose, hypotaurine, necrostatin-1, melatonin	↑ Tissue viability ↓ Apoptosis
Zhu et al., 2021 [104]	Bovine	Slow-freezing; trehalose and KSR; uncontrolled slow-freezing	↓ Apoptosis ↑ Cell viability Preservation of structural integrity and seminiferous epithelial cohesion Maintenance of SSCs germline characteristics

↑ Increase; ↓ Decrease.

**Table 5 biology-13-00542-t005:** Potential methods of mitigating IRI in OT fragments.

Author, Date	Model	Treatment	Main Findings
He et al., 2017 [109]	Mouse	K252a Rapamycin	↓ Primordial follicle activation
Eken et al., 2019 [116]	Rat	Etanercept	↑ GSH and SOD levels ↓ Inflammation and apoptosis
Manavella et al., 2019 [123]	Mouse	ASCs	↑ Vessel density
Celik et al., 2020 [56]	Rat	Rapamycin	↓ Primordial follicle activation
Liu et al., 2021 [111]	Mouse	Rapamycin	↓ Primordial follicle activation ↑ Ovarian survival rate ↓ Apoptosis
Olesen et al., 2021 [113]	Human	NAC	↑ Expressions of SOD1, HMOX1, and CAT ↓ IRI ↓ Follicle apoptosis ↑ Follicle density ↓ Expression of VEGFA
Li et al., 2021 [115]	Human	GSH, UTI, or GSH+UTI	↑ Follicle survival ↑ Antioxidant enzyme activity ↑ Angiogenesis ↓ Oxidative stress ↓ Inflammation
Ahmadi et al., 2021 [107]	Mouse	Taurine	Prevention against oxidative stress ↑ Angiogenesis ↓ Apoptosis ↑ Follicle survival and growth
Sanamiri et al., 2022 [117]	Mouse	L-carnitine	↑ Number of follicles ↑ Estradiol and progesterone production ↓IL-6, TNF-α and MDA levels
Rodrigues et al., 2023 [106]	Mouse	Erythropoietin	↑ Follicle viability ↓ Follicle degeneration ↑ Angiogenesis ↓ Fibrotic areas
Bindels et al., 2023 [112]	Mouse	Rapamycin or LY294002	↓ Follicle proliferation Maintenance of primordial follicle reserve
Celik et al., 2023 [125]	Mouse	Anti-Mullerian Hormone	↓ Primordial follicle loss
Ebrahimi & Nasiri, 2024 [114]	Mouse	Estradiol and NAC	↑ Primordial, preantral, and antral follicle numbers ↓ Levels of TNF-α and FGF-2 ↑ Levels of IL-1β and IL-6 ↑ Levels of VEGF

↑ Increase; ↓ Decrease.

## Data Availability

Data sharing is not applicable.

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
