# Peer review of "The Low Survivability of Transplanted Gonadal Grafts: The Impact of Cryopreservation and Transplantation Conditions on Mitochondrial Function"

_biology, 2024, doi:10.3390/biology13070542_

Round 1

Reviewer 1 Report

Comments and Suggestions for Authors

The authors summarized the limitations of current gonadal tissue preservation techniques with a focus on the mitochondrial damages caused by cryopreservation. They discussed the freezing conditions and cryoprotectants, and their impacts on mitochondrial function, oxidative stress, and ischemia-reperfusion damage. The discussion was followed by preventing strategies taken in the field, yet with both success and limitations. While this article provides valuable insights for future studies in improving cryopreservation techniques for gonadal grafts by targeting mitochondrial damage, the authors should work further on the language and improving the specificity of their expressions before the manuscript is accepted.

For improving specificity and reducing vague expressions, an example is:

Line 159:  The description of “reduces the cells’ antioxidant defenses” is vague. What are the pathways that are downregulated or upregulated? For example, are glutathione levels decreased?  Is the expression of SOD2 or GPX4 decreased? Depending on the antioxidant pathways, the strategies for reducing the related stress might be disparate and therefore important to be discussed.

Comments on the Quality of English Language

For improving clarity and certain spellings, examples are (and not limited to):

Line 141-142. Cluttered sentence.

Line 242. Should be “ischemia”.

Reviewer 2 Report

Comments and Suggestions for Authors

Dear Authors,

The manuscript is reviewed. It is well-written, organized, and novel in information. I have a few suggestions for improving the present form of the manuscript. 

1. Provide a flow chart or figure depicting the potential implications of cryopreservation-associated damage to the gonadal tissues for the better understanding of readers.

2. A flow chart or figure showing the target of free radicals and the mitochondria. 

3. A flow chart depicting the possible targets of cryopreservation and its implications in the future. 

best
